Exploring the classical and numerical Delboeuf illusion: the impact of transcranial alternating current stimulation on magnitude processing

Santacà Maria maria.santaca@univie.ac.at santacamaria@gmail.com 1
Carabba Francesco 2
Fernando Achini Mihindukulasooriya 2
Pitocco Enrico 2
Battaglini Luca 2
Granziol Umberto 2
1 Department of Behavioural and Cognitive Biology, University of Vienna , Vienna , Austria
2 Department of General Psychology, University of Padua , Padua , Italy
Prpic Valter
Electronic publication date: 2025 Mar 10
Publication date: 2025
Volume: 13
Electronic Location ID: e19064
Received 2024 Dec 20; Accepted 2025 Feb 6
Copyright: ©2025 Santacà et al.
Copyright year: 2025
Copyright holder: Santacà et al.
License: This is an open access article distributed under the terms of the Creative Commons Attribution License, which permits unrestricted use, distribution, reproduction and adaptation in any medium and for any purpose provided that it is properly attributed. For attribution, the original author(s), title, publication source (PeerJ) and either DOI or URL of the article must be cited.
License URL: https://creativecommons.org/licenses/by/4.0/

Keywords: Brain stimulation, Human perception, Magnitude processing, Numericaldiscrimination, Spatial discrimination

Funding: The Austrian Science Fund (FWF) through an ESPRIT grant 10.55776/ESP433 This work was supported by the Austrian Science Fund (FWF) through an ESPRIT grant (Grant DOI: 10.55776/ESP433) to Maria Santacà. The funders had no role in study design, data collection and analysis, decision to publish, or preparation of the manuscript.

==============================
Understanding cognitive and neural mechanisms underlying quantity processing is crucial for unraveling human cognition. The existence of a single magnitude system, encompassing non-symbolic number estimation alongside other magnitudes like time and space, is still highly debated since clear evidence is limited. Recent research examined whether spatial biases also influence numerosity judgments, using visual illusions like the Delboeuf illusion. While findings support a generalized magnitude system, direct comparisons of spatial and numerical Delboeuf illusions are missing. This study explored whether perceptual biases similarly affect different magnitude processing and whether transcranial alternating current stimulation (tACS) modulates these processes. Participants underwent three tACS conditions (seven Hz, 18 Hz, placebo) while performing tasks involving the classic and numerical Delboeuf illusions. We hypothesized that theta-frequency tACS (seven Hz) would enhance visual integration and illusion strength, while beta tACS (18 Hz) would reduce it by promoting visual segregation. Results indicated higher discrimination accuracy in area-based tasks than numerical judgments. Nonetheless, a significant correlation between performances in spatial and numerical illusions supported the existence of a shared mechanism for magnitude processing. Contrary to expectations, seven Hz tACS reduced the perceptual illusion’s strength. No significant interaction emerged between tACS frequency and discrimination abilities. These findings deepen our understanding of the cognitive processes involved in magnitude perception, potentially supporting the hypothesis of a generalized magnitude system. They also highlight the potential and limitations of non-invasive brain stimulation techniques, such as tACS, in modulating perceptual processes, offering insights into the neural underpinnings of quantity perception.

Introduction

Quantitative discrimination, namely the cognitive process involved in distinguishing numerical quantities, is a foundational aspect of human cognition with broad implications across various domains. The ability to perceive and manipulate quantities underpins human interactions with the environment and informs higher-order cognitive functions. This is valid not only for “rudimentary” functions, such as enumerating objects and comparing sizes and amounts, but also for more complex ones, such as decision-making processes involving mathematical reasoning and analysis.

Understanding what influences numerosity estimation is hampered by the intrinsic covariance of numerosity in natural environments with several other physical properties, including the size, density, and area of the item configuration. Humans use a process known as the approximate number system (ANS) to estimate quantities with no upper limit. However, Weber’s Law states that its accuracy varies with ratio (e.g., Gallistel & Gelman, 2000; Halberda et al., 2012; Izard et al., 2009; Walsh, 2003): distinguishing between 30 and 60 items, which have a ratio of 0.50, is simpler than between 30 and 45 items, which have a ratio of 0.67. The ANS might also control the processing of continuous quantities, including time and space, in addition to discrete numbers (numerosity). This led to hypothesize the existence of a shared cognitive mechanism that encodes non-symbolic number estimation alongside other magnitudes, such as time and space (Gallistel & Gelman, 2000; Walsh, 2003; Gallistel, 1990). Supporting empirical evidence comes from behavioral studies showing that perceptual judgments of numerosity can be influenced by variations in unrelated continuous quantities (e.g., De Hevia et al., 2008; Kaufmann et al., 2005). For example, Lourenco & Longo (2010) demonstrated a cross-dimensional transfer in associative learning in 9-month-old infants. Infants’ expectations in the task with one dimension (i.e., size) was influenced by the information related to other magnitude dimensions (i.e., numerosity; duration): nine-month-olds anticipated that numerosity and duration would follow the same color-pattern mapping after learning that larger items had a particular color-pattern. These findings provide support to the hypothesis of a shared magnitude system wherein space, numerosity and time are represented in an abstract way not tied to specific dimensions. Nevertheless, there is still ongoing debate regarding the existence of a shared magnitude system since other researchers reported discrepancies in spatial, temporal, and numerical processing (Dormal, Andres & Pesenti, 2008; Droit-Volet, Clément & Fayol, 2008).

To shed light on this aspect, recent studies exploited visual illusions to assess whether perceptual biases that influence spatial decisions also affect numerosity judgements. If the same cognitive system encodes both spatial and numerical ability, perceptual biases should impact judgments about numerosity and continuous quantities similarly. In the Müller-Lyer illusion, where two lines of equal length appear different, human participants were found to be influenced by the perceived length of dot arrays when comparing numerosities (Dormal et al., 2018). Similarly, the horizontal-vertical illusion, where a vertical line is perceived longer than the same line placed horizontally, produced similar effects on numerosity judgments (Pecunioso, Miletto Petrazzini & Agrillo, 2020). Additionally, researchers have explored the relationship between area size and numerosity using the Ebbinghaus illusion. This illusion involves observers overestimating the size of a circle surrounded by smaller circles and underestimating the size of the same circle when surrounded by larger circles. Human participants tended to overestimate the number of dots presented in the perceived larger area, according to this illusion (Picon, Dramkin & Odic, 2019). However, this study faced challenges in controlling low-level features, such as dot size and spacing, which could confound the results. In contrast, Adriano, Girelli & Rinaldi (2022) provided more robust evidence using the Ponzo illusion (where the perceived size of an object is influenced by converging lines that create a false sense of depth) and the connectedness illusion (where connecting two or more items with a visual element alters their perceived numerosity), while controlling for low-level features. Their findings demonstrate that numerosity perception is influenced by spatial illusions, supporting the hypothesis of a shared processing mechanism for numerosity and spatial dimensions in the parietal cortex (Walsh, 2003).

Following this evidence, in a previous study, we examined whether the same perceptual biases underlying the Delboeuf illusion can be also observed in numerical estimation tasks (Santacà & Granziol, 2023). Humans were instructed to select a target numerosity (larger or smaller) of squares between two sets that differed in their numerosity. Participants were also presented with illusory trials in which the same numerosity was presented in two different contexts, against a large and a small background, resembling the Delboeuf illusion (Fig. 1A). In these illusory trials, humans demonstrated numerical biases in agreement with the perception of the classical version of the Delboeuf illusion, with the array perceived as larger appearing more numerous. These findings suggest a potential interconnection between spatial perception and numerical cognition in humans. Indeed, our results support the hypothesis of a single magnitude system, as perceptual biases that influence spatial decisions seem to affect numerosity judgements in the same way (Santacà & Granziol, 2023). Nonetheless, in that previous study, participants were not tested with the classical Delboeuf illusion, in which the same area is presented in the two contexts (Fig. 1B). Therefore, it was not possible to ascertain whether there was a correlation between the perception of the classical Delboeuf illusion (i.e., resembled with areas) and the numerical version. This leaves open the question of whether the Delboeuf illusory effect is of the same magnitude with both types of quantities and, thus, if this perceptual bias influences spatial decisions and numerosity judgements in the same way. This highlights the need for further investigation to elucidate the intricacies of perceptual phenomena and their interplay with cognitive processes.

Figure 1 The classical and numerical Delboeuf illusion.

The Delboeuf illusion is a perceptual phenomenon where two objects of equal size are perceived differently depending on their surrounding context. The figure illustrates both a numerical version of the Delboeuf illusion (A) and the classical version (B), where respectively the same numerosity or the same target area is surrounded by two different contexts. Humans proved to typically underestimate the numerosity and the target area encompassed by the larger ring, and instead tend to overestimate the numerosity and the target area encompassed by the smaller ring.

In this study, we aim to investigate this phenomenon not only at a behavioral level but also at a neural level. Indeed, we tested participants across three experimental sessions, each involving a different neural stimulation condition: theta-band stimulation (seven Hz), beta-band stimulation (18 Hz), and a no-stimulation (sham) control condition. During each session, participants were presented with both the classical and numerical versions of the Delboeuf illusion. This design allows us to investigate how specific neural oscillations, induced via transcranial alternating current stimulation (tACS), influence perceptual and cognitive processes involved in magnitude processing. tACS has emerged as a promising tool for exploring these mechanisms (Wischnewski, Alekseichuk & Opitz, 2023), as it allows researchers to modulate brain activity by applying alternating currents that entrain (i.e., align neural oscillatory rhythms with the frequency imposed by electrical stimulation) specific neural oscillations, potentially influencing cognitive processes. For instance, Battaglini et al. (2020) investigated the effects of tACS on visual crowding at specific frequencies targeted to parietal regions. Visual crowding, which defines a basic boundary for conscious perception, refers to the inability to distinguish items when flankers are close by Levi (2008). Previous studies have linked crowding to beta oscillations (12–30 Hz) in the parietal cortex, where greater beta amplitude was associated with better crowding resilience (Ronconi, Bertoni & Marotti, 2016; Ronconi & Bellacosa Marotti, 2017). Supporting this, Battaglini et al. (2020) found that beta-frequency tACS stimulation on the parietal region enhanced perceptual segmentation of targets from distractors in crowding conditions. The importance of the parietal areas in crowding tasks was further supported by Di Dona et al. (2024). Additionally, Stonkus et al. (2016) revealed that theta-frequency tACS stimulation (i.e., seven Hz) improved performance in a perceptual integration task. By comparing performances across different types of magnitude discrimination tasks under varying tACS conditions, we aim to provide a more comprehensive understanding of the perceptual and cognitive processes involved in magnitude processing. Modulating the strength of the Delboeuf illusion using different tACS stimulation frequencies would allow us to determine whether these modulations affect both types of illusions (classical and numerical) similarly and to the same extent. Such findings would provide further support for the hypothesis of a generalized magnitude system in humans. The Delboeuf illusion involves processing both the central element (or squares in the numerical version of the illusion) and the surrounding background. We hypothesize that tACS at specific frequencies may modulate the perceptual ability to integrate or segregate elements in an image, thus affecting the strength of the illusion. Specifically, the choice of tACS frequencies at seven Hz (theta band) and 18 Hz (beta band) is based on evidence linking these oscillations to perceptual and visual integration processes. Studies have shown that theta oscillations are involved in integrating visual elements into a global percept (Stonkus et al., 2016; Wutz et al., 2016), while beta oscillations are associated with visual element segregation (Battaglini et al., 2020; Di Dona et al., 2024; Ghiani et al., 2021; Ronconi & Melcher, 2017). In particular, seven Hz tACS over the parietal cortex has been shown to enhance performance in perceptual integration tasks (Stonkus et al., 2016), whereas beta stimulation facilitates the segregation of visual elements under crowding conditions (Battaglini et al., 2020; Di Dona et al., 2024). Based on these findings, we hypothesize that seven Hz tACS over the parietal region will enhance the integration of visual elements into a unified percept, thereby strengthening the Delboeuf illusion compared to the no-stimulation (sham) condition. Conversely, 18 Hz tACS over the right and left parietal regions is expected to increase the segregation of visual elements, reducing the strength of the illusion compared to sham. To summarize, modulating the strength of the Delboeuf illusion using tACS at seven Hz and 18 Hz allows us to explore the underlying perceptual and cognitive processes. If both classical and numerical forms of the illusion are similarly affected, this would support the hypothesis of a generalized magnitude system in humans. Finally, we performed a power simulation study to estimate the sample size required to achieve specific statistical power in future research, based on the results of the present study.

Materials & Methods

Ethics statement

All procedures involving human participants were in accordance with the ethical standards of the institutional committee and with the 1964 Helsinki Declaration and its later amendments or comparable ethical standards. The study was approved on the 11th of April 2023 by the ethics committee (Protocol no. 5179, code no. 766656C6F2B60F2D3731E72418CD558B) of the Department of General Psychology at the University of Padova (Italy). Data collection was done between 23 May 2023 and 24 July 2023.

Participants

For this study, we recruited 48 adult volunteers (38 females, 10 males; mean age ± SD = 24.19 ± 3.02 years). Written informed consent was obtained from all participants, all of whom were students of the University of Padova pursuing either a bachelor’s or master’s degree. Such sample size was chosen to align with the previous work of Santacà & Granziol (2023), which is the only prior study with a similar design and variables. All participants reported having normal or corrected-to-normal vision. Eligibility for transcranial electrical stimulation was assessed through a pre-screening test conducted at the beginning of each session. Exclusion criteria for the study include neurological disorders affecting visual and/or numerical abilities, substance abuse, and any medical condition posing a risk to participants (e.g., pacemaker, epilepsy, migraine auras). Not all participants met the criteria for receiving tACS so they were excluded. Consequently, the final sample consisted of 34 participants (29 females, five males; mean age ± SD = 24 ± 2.7 years). A post-stimulation questionnaire was administered at the end of each session to ensure participants well-being and to evaluate whether they anticipated receiving true stimulation or the sham condition (i.e., placebo; see Fertonani, Ferrari & Miniussi, 2015).

Experimental set-up and tACS stimulation

The study was conducted in a dimly lit room to minimise the detection of tES-induced phosphenes (Evans, Palmisano & Croft, 2022). Participants were instructed to use a chin rest positioned 57 cm away from the monitor. The stimuli were displayed on an LCD ASUS monitor with a resolution of 1, 280  × 1,024 and a refresh rate of 60 Hz. tACS was administered over the parietal areas using a BrainSTIM device (E.M.S. srl) at an intensity of one mA. Carbonised rubber electrodes, measuring 5  × 5 cm and covered in sponges, were placed at the locations corresponding to P3 and P4 on a 64-channel EEG cap arranged according to the international 10–20 system. In the real stimulations, the current was applied for 45 min (with 10 s fade-in and -out periods at the stimulation’s beginning and end). In the sham tACS, the current was turned off 10 s after the beginning of the stimulation (with 10-s fade-in and –out periods, for a total of 30 s). Most participants reported experiencing either a low level or absence of fatigue during the test, independently of the stimulation type (χ29 = 8.14, p = 0.52). No phosphenes were reported. Moreover, they were not able to discriminate between real stimulations and placebo, as confirmed by a Pearson’s Chi-Square test for frequency distribution (χ21 = 1.76, p = 0.18; the observed probability of reported ‘stimulation present’ was 0.60 in the seven Hz tACS condition, 0.52 in the 18 Hz tACS condition and 0.75 in the sham condition). Stimulation parameters were selected in accordance with the safety guidelines provided by Antal et al. (2017). Information regarding the electric field distribution (magE, in V/m) can be found in Fig. S1.

Stimuli

We used two distinct types of stimuli (Fig. 2). For the numerical discrimination, the stimuli consisted of two arrays of orange squares set within two white circular backgrounds placed into two black rectangles. On the other hand, for the continuous quantity discrimination, the stimuli consisted of two orange circles within two white circular backgrounds, which were further enclosed within two 4.5 × 4.5 cm black rectangles as for the other discrimination.

Figure 2 Examples of experimental stimuli.

The figure shows an example of the four types of control trials for both discriminations: a large trial (A), a small trial (B), a congruent trial (C) and an incongruent trial (D). The figure also shows an example of illusory trial (E), in which the same numerosity/same target circle is presented in two different-sized backgrounds resembling the Delboeuf illusion.

For each discrimination, we arranged two different types of trials: control trials and illusory trials. In control trials, there was a real difference between the two stimuli: 10 squares vs. 12 squares for the numerical discrimination (i.e., ratio 0.83) whereas, for the continuous quantity discrimination, the circles’ area differed by the same ratio proposed in the numerical discrimination. Regarding control trials, four different combinations of numerosity/circles and background were used as in a previous study (Santacà & Granziol, 2023). In ‘large trials’, the two target stimuli to discriminate were presented in two identical large backgrounds (4.22 cm in diameter; Fig. 2A). Conversely, in ‘small trials’, the two target stimuli were presented in two identical small backgrounds (2.79 cm in diameter; Fig. 2B). In the remaining two types of trials, different backgrounds were utilized within each pair of stimuli. In ‘congruent trials’, the larger target stimulus was presented in the large background and the smaller stimulus in the small background (2.79 cm and 4.22 cm in diameter; Fig. 2C). In ‘incongruent trials’, the large background included the smaller stimulus, and the small background surrounded the larger stimulus (2.79 cm and 4.22 cm in diameter; Fig. 2D). Lastly, illusory trials consisted of the same numerosity or the same circles in two different backgrounds, a large and a small one (2.79 cm and 4.22 cm in diameter; Fig. 2E), resembling the numerical or classical Delboeuf illusion, respectively. For the numerical discrimination, overall, we arranged six different pairs of each type of control trials and also illusory trials in which both the position and the size of the squares changed. The squares’ sides ranged between 0.15 cm and 0.30 cm. Similary, the continuous quantity discrimination involved six different pairs of each type of control trials and also illusory trials in with the circles’ diameters ranging between 1.64 cm and 2.35 cm.

As reported also in the previous study (Santacà & Granziol, 2023), one problem when assessing numerical discrimination is that numerosity naturally covaries with other physical properties, namely non-numerical continuous variables (e.g., the total area of all elements of a stimulus, the size of the elements, the overall space encompassed by the elements, and their density). Thus, to ensure that participants did not base their decisions on these other physical attributes, we controlled these variables in the control trials, and we presented the same numerosity in illusory trials. Specifically, we controlled the stimuli pairings based on their cumulative surface area. We deliberately avoided having the exact same cumulative surface areas for the two stimuli: it would generally result in more frequent smaller-than-average square stimuli in larger numerosities, information that could lead to biased responses. Instead, we varied the ratio of cumulative surface area between smaller and larger numerosities. One-third of stimuli had a ratio between 76% and 85%, another third between 86% and 95%, and the remaining third between 96% and 105%. Furthermore, density and space taken by the squares are inversely correlated. Therefore, these two variables were accounted for as half of the stimuli pairing was controlled for density and the other for space occupied by the squares. Consequently, the results of illusory trials would exclusively reflect the direct effect of the Delboeuf illusion on numerosity estimation, rather than the indirect impact of a biased perception due to other physical attributes. Moreover, we also rotated each numerosity array in each illusory trial to minimize the risk that participants would notice that the squares in the two illusory arrays were identical in size and location.

Procedure

Each experimental session consisted of two blocks: one featuring the numerical stimuli and the other one featuring the other type of stimuli (continuous quantity, namely circles). Each block had five trial types and lasted approx. 20 min, with a 5-minute interval, resulting in a total duration of 45 min per experimental session. Considering the three different types of stimulation, the study entailed 30 conditions per participant, each consisting of 60 trials for a total of 1,800 trials. Both oral and written instructions were provided before starting the experiment. During the experimental session, participants were presented with pairs of stimuli of the same type (numerical or continuous quantity discrimination) placed eight cm apart. Using a QWERTY keyboard, they were asked to indicate which stimulus appeared to be more numerous or larger in size depending on the discrimination. More specifically, participants used the “S” button to choose the stimulus on the left side of the screen and the “L” button for the one on the right side. The order in which the two blocks were presented, such as the different trial types within each block, was randomized across participants.

At the beginning of each block, a fixation cross appeared at the center of the screen for 250 ms. Participants were directed to maintain their gaze on the center of the monitor screen throughout each block. Paired stimuli were displayed for 150 ms, a duration too brief to allow for saccadic eye movements towards the target position or verbal counting during numerical discrimination. Following the stimulus presentation, a white screen was shown for 550 ms, during which participants made their choice. If they did not provide their response within this timeframe, the trial was considered null. Overall, 2.43% of all the trials in the numerical discrimination (large trials: 2.60%, small trials: 2.04%, congruent trials: 2.14%, incongruent trials: 2.32%, Delboeuf illusion: 3.06%), and 2.34% in the continuous quantity discrimination (large trials: 3.07%, small trials: 2.37%, congruent trials: 2.17%, incongruent trials: 1.99%, Delboeuf illusion: 2.22%), were excluded due to this criterion.

The experiment consisted of three sessions for each participant, conducted on non-consecutive days at least 48-h apart. In each session, participants performed both discriminations, but they were subjected to different stimulation conditions. More specifically, participants were subjected to the sham condition, where no stimulation was administered, along with two other conditions involving stimulation at different frequencies: seven Hz and 18 Hz tACS conditions. The assignment of the different stimulation conditions was randomized across the sessions, ensuring that each participant experienced all conditions in a different order.

Statistical analyses

Data were analyzed in R version 3.5.2 (R Core Team, 2016). For all three stimulations (seven Hz, 18 Hz, sham), we recorded accuracy in terms of selecting the larger target stimulus and numerosity for control trials. In illusory trials, we scored as ‘correct’ the choices for the stimulus and the numerosity presented in the small context. At the individual level, we used binomial tests to compare the choices for the larger target stimulus and numerosity in control trials and for the stimulus and the numerosity presented in the small context in illusory trials (chance level = 0.5). We performed group analyses on the frequency of choices for the larger target stimulus and numerosity in control trials and for the stimulus and the numerosity presented in the small context in illusory trials. Not all data were normally distributed (Shapiro–Wilk test, p < 0.05); thus, we performed one-sample t tests or Wilcoxon-signed rank tests (chance level = 0.5). Cohen’s d effect size was reported for all one-sample in the Table S2, (cohensD() function of the lsr R package (Navarro, 2015)).

Considering only the sham condition, we conducted a Pearson (normal distribution)/Spearman correlation test to evaluate the relationship between performance on the numerical discrimination and continuous quantity discrimination tasks. This analysis was split into two parts: one part included control trials (small and large trials) where the Delboeuf illusion was not expected to occur due to identical background sizes; the other part included trials where the illusion was likely to manifest (congruent, incongruent, and illusory trials), given the differing background sizes. For each correlation tested, each data point (for each participant), was the average of the performances across each type of trial.

We also assessed the accuracy of responses by fitting a generalized mixed-effects model for binomial distributions (GLMM) with three variables: the stimulation (seven Hz, 18 Hz or sham), the discrimination (numerical or continuous quantity), and the stimulus type (large, small, congruent, incongruent or illusory trials). We fitted each of these variables, as well as their two- and three-way interactions, as fixed effects whereas we fitted subjects as clustering variable and random factor (i.e., random intercept model). Sum contrasts were set for the three abovementioned predictors. GLMMs were estimated with a Maximum Likelihood (Laplace Approximation) procedure with the function glmer() from the lme4 package (Bates et al., 2008). Omnibus tests for each main effect was estimated by using the Anova (type=“III”) function of the car package was used (Fox et al., 2012). Whenever an effect emerged as statistically significant, post-hoc comparisons were performed with the function emmeans() from the emmeans package (Lenth, 2024). Considering the number of comparisons that could arise from high-order effects such as interactions, not all the comparisons were analyzed, since some of them were not of interest. The target post-hoc comparisons always referred to the difference among levels in a former variable across specific levels of a latter variable. For instance, a comparison could focus on the difference between congruent vs incongruent trials for continuous discrimination; the same comparisons were then tested also for numerical discrimination (see Table S2). In this way, it was reduced the chance of committing Type I error due to comparisons that were beyond the aim of the present work. Nonetheless, the False Discovery rate method (Benjamini & Hochberg, 1995) was used to adjust post-hoc comparisons. For each comparison, Odds Ratios (ORs), their 95% confidence intervals (CI), statistics (z), standard error (SE), p-values (p) are also reported. As suggested by several works (Harris, 2021; Lenzi et al., 2015), when reporting OR, the outcomes may be presented in two different formats: as a percentage difference in likelihood, which is calculated by subtracting the Odds Ratios from 1.0, and as “n times less/more likely”, which is determined by dividing 1.0 by the OR in the former case (i.e., “less”). In the present paper, the latter way was preferred, since ORs below 1.0 may be less straightforward and intuitive for interpreting the strength of associations compared to ORs above 1.0. Overall, three different GLMMs were performed: two models tested the interaction between the stimulus type and the stimulation only in the numerical discrimination or only in the continuous quantity discrimination. Furthermore, we performed an overall model including the discrimination type with the two former predictors. Moreover, potential differences in reaction times among stimuli were tested through a GLMM. In particular, stimulus type was set as fixed factor, while the subject ID was set as random factor. In case of a statistically significant effect, post-hoc comparisons were performed.

For all the tests/comparisons that referred to a GLMM, we used OR also as an effect size measure (Szumilas, 2010; Bland, 2000; Chen, Cohen & Chen, 2010). For all the other tests, we used Cohen’s d. For the χ2 related to the main effect and obtained via the Anova() function, effects size were not calculated, since χ2 here represents an unfocused omnibus test useful to comment specific comparisons and their OR (Rosenberg, 2010).

Lastly, in order to enhance more accurate results and replications of the present study, a sample size simulation and the related power analysis was performed using the parameters of the GLMM on the overall model. This analysis also allows us to predict the sample size needed to have sufficient power. In particular, the interaction effect between the Stimulus factor and the Stimulation factor was retested. The predicted effect size for this effect, which was used for the power analysis, is the same identified in this study. The alpha was kept at 0.05. The R package simr (Green & MacLeod, 2016) was used, since allows to calculate power for generalized linear mixed models from the lme4 package. These power calculations were performed using Monte Carlo simulations. Specifically, the powercurve() function to generate power curves was used, to evaluate the trade-offs between power and sample size. Basically, a new set of data was simulated based on the provided fitted model; then the model was refitted to this newly simulated data, and a statistical test on the refitted model was conducted. The test would either successfully detect the effect or make a Type II error by failing to detect the effect. This flow was replicated 1,000 times. The power of the test was then determined as the proportion of times the effect is successfully detected in step three.

Results

Behavioral level: sham condition

Numerical discrimination

For the numerical discrimination, individual analyses revealed that 23 out of 34 participants selected the larger numerosity significantly more than chance in control trials (mean ± SD = 61.00 ± 9.16%; Table S1). Considering the Delboeuf illusory trials, nine out of 34 participants selected the numerosity presented in the small context significantly more than chance whereas interestingly, five selected the one presented in the large context more often than chance (mean ± SD = 54.51 ± 16.51%; Table S1).

Group analyses revealed that participants selected the larger numerosity significantly more than chance in control trials (mean ± SD = 62.35 ± 8.30%; Wilcoxon-signed rank test, Z = 0.98, p < 0.001; specific analyses for each type of control trials are reported in the Table S2). Overall, participants did not perceive the numerical Delboeuf illusion, since they did not select any numerosity significantly more than chance (mean ± SD = 54.51 ± 16.85%; one-sample t test, t 33 = 1.56, p = 0.128).

Continuous quantity discrimination

Individual analyses revealed that 23 out of 34 participants selected the larger target stimulus significantly more than chance in control trials (Table S1). Considering the Delboeuf illusory trials, 27 out of 34 participants selected the stimulus presented in the small context significantly more than chance (Table S1).

Group analyses revealed that participants selected the larger target stimulus significantly more than chance in control trials (mean ± SD = 62.35 ± 8.30%; Wilcoxon–signed rank test, Z = 1.07, p < 0.001; specific analyses for each type of control trials are reported in the Table S2). Participants responded to the Delboeuf illusion as expected, selecting the stimulus presented in the small context significantly more than chance (mean ± SD = 80.38 ± 17.70%; Z = 1.10, p < 0.001).

Correlation between numerical and continuous quantity discrimination

Considering only those control trials in which the Delboeuf illusion has no effect (large and small trials), we found a significant correlation between performance in the two discriminations (Pearson correlation; r32 = 0.37, p = 0.03; Fig. 3). Even considering only those trials in which the illusion has an effect (congruent, incongruent, and illusory trials), we found a significant correlation between performance in the two discriminations (Spearman correlation; r32 = 0.34, p = 0.0497; Fig. 4). The first correlation suggests that participants who showed higher discrimination ability with continuous quantities also better discriminate between different numerosities. The second correlation, instead, suggests that the participants that are more influenced by the Delboeuf illusion when it is resembled with a continuous quantity, are also more influenced by it when it is resembled with numerosity arrays.

Figure 3 Correlation between numerical and continuous quantity discrimination.

The correlation considers only small and large trials in which the Delboeuf illusion has no effect due to the identical background. The x-axis represents continuous quantity discrimination, and the y-axis represents numerical quantity discrimination. Each point corresponds to a subject, and the trend line shows the relationship between these two discriminations.

Figure 4 Correlation between numerical and continuous quantity discrimination.

The correlation considers those trials (congruent, incongruent, and illusory trials) in which the Delboeuf illusion has an effect due to use of the two different backgrounds. The x-axis represents continuous quantity discrimination, and the y-axis represents numerical discrimination, with points indicating individual performances and a trend line highlighting the relationship under illusion conditions.

Neural level: seven Hz and 18 Hz tACS stimulations

Numerical discrimination

In the seven Hz tACS stimulation, individual analyses revealed that 17 out of 34 participants selected the larger numerosity significantly more than chance in control trials (Table S1). Considering the Delboeuf illusory trials, 12 out of 34 participants selected the numerosity presented in the small context significantly more than chance whereas, interestingly, four selected more than chance the one presented in the large context (Table S1).

Group analyses revealed that, in the seven Hz tACS stimulation, participants selected the larger numerosity significantly more than chance in control trials (mean ± SD = 58.88 ± 8.58%; one-sample t test, t33 = 6.21, p < 0.001; specific analyses for each type of control trials and the related figure are reported in the Supplemental Information). Overall, participants seemed to perceive the numerical Delboeuf illusion, as shown by the selection of the numerosity presented in the small context significantly more than chance (mean ± SD = 57.88 ± 18.88%; t33 = 2.43, p = 0.021*).

In the 18 Hz tACS stimulation, individual analyses revealed that 18 out of 34 participants selected the larger numerosity significantly more than chance in control trials (Table S1). Considering the Delboeuf illusory trials, 10 out of 34 participants selected the numerosity presented in the small context significantly more than chance whereas, interestingly, four selected more than chance the one presented in the large context (Table S1).

Group analyses revealed that, in the 18 Hz tACS stimulation, participants selected the larger numerosity significantly more than chance in control trials (mean ± SD = 59.59 ± 8.69%; one-sample t test, t33 = 6.289, p < 0.001; specific analyses for each type of control trials and the related figure are reported in the Supplemental Information). Overall, participants did not perceive the numerical Delboeuf illusion, as they did not select any numerosity significantly more than chance (mean ± SD = 55.55 ± 16.59%; t33 = 1.951, p = 0.060).

Considering the GLMM, it was not observed that the stimulation had a statistically significant effect on the participants’ accuracy (χ22 = 4.11, p = 0.128). However, a statistically significant effect of the trial type was found (χ24 = 248.58, p < 0.001). Specifically, participants reported a significantly lower accuracy in congruent trials, compared to the other trial type (all p < 0.001). In the case of incongruent trials, the accuracy was significantly higher, compared to the other trial type (all p < 0.05). The accuracy was also higher in large trials, especially when compared to both small (OR = 1.158; p < 0.001) and illusory trials (OR = 1.353; p < 0.001). The accuracy decreased in small trials compared to illusory trials (OR = 1.169; p < .001). The complete list of these post-hoc comparisons, including ORs, their 95% CI, z-values, SE, and p-values can be found in Table S2. Lastly, considering the interaction between the stimulation and the trial type, no statistically significant effect was found (χ28 = 9.16, p = 0.329).

Continuous quantity discrimination

In the seven Hz stimulation, individual analyses revealed that 26 out of 34 participants selected the larger target stimulus significantly more than chance in control trials (Table S1). Considering the Delboeuf illusory trials, 29 out of 34 participants selected the stimulus presented in the small context significantly more than chance (Table S1).

Group analyses revealed that, under the seven Hz tACS stimulation, participants selected the larger target stimulus significantly more than chance in control trials (mean ± SD = 61.58 ± 7.40%; Wilcoxon-signed rank test, Z = 1.08, p < 0.001; specific analyses for each type of control trials and the related figure are reported in the Supplemental Information). Participants also showed to perceive the Delboeuf illusion, by selecting the stimulus presented in the small context significantly more than chance (mean ± SD = 78.81 ± 18.44%; Z = 1.07, p < 0.001).

In the 18 Hz tACS stimulation, individual analyses revealed that 28 out of 34 participants selected the larger target stimulus significantly more than chance in control trials (Table S1). Considering the Delboeuf illusory trials, 29 out of 34 participants selected the stimulus presented in the small context significantly more than chance (Table S1).

Group analyses revealed that, in the 18 Hz tACS stimulation, participants selected the larger target stimulus significantly more than chance in control trials (mean ± SD = 63.00 ± 6.98%; Z = 1.159, p < 0.001; specific analyses for each type of control trials are reported in the Supplemental Information). The Delboeuf illusion was perceived by the participants as they selected the stimulus presented in the small context significantly more than chance (mean ± SD = 82.05 ± 17.93%; Z = 1.097, p < 0.001).

Considering the GLMM, a statistically significant effect of the stimulation on the participants’ accuracy was found (χ22 = 9.52, p = 0.009): participants were more likely to correctly respond when subjected to 18 Hz tACS stimulation, compared to the seven Hz tACS stimulation (OR = 1.111; p < 0.01). However, no differences in accuracy emerged when comparing both 18 Hz and 7 Hz tACS stimulations with the sham condition (OR =1.061−0.955; p > 0.05). Concerning the trial type, a statistically significant effect was observed (χ24 = 4437.38, p < 0.001). In detail, compared to all the other trials, the accuracy on congruent trials was lower (all p < 0.001), as also found for the numerical discrimination. With incongruent trials, the accuracy was higher (all p < 0.001), compared to all the other trial types as also found for the numerical discrimination. The performances in both large and small trials were not significantly different (OR = 1.043; p = 0.273). Instead, in illusory trials, the accuracy was significantly higher, compared to both large (OR = 0.508; p < 0.001) and small trials (OR = 0.486; p < 0.001). The complete list of these post-hoc comparisons, including ORs, their 95% CI, z-values, SE, and p-values can be found in Table S2. Lastly, considering the interaction between the stimulation and the trial type, no statistically significant effect was found (χ28 = 5.78, p = 0.672).

Comparison between numerical and continuous quantity discrimination

In the overall model, the effect of the discrimination emerged as statistically significant (χ21 = 443.75, p < 0.001). In particular, participants were significantly more accurate in the continuous quantity discrimination than the numerical one (OR = 1.469; p < 0.001, Fig. 5). Furthermore, a statistically significant effect of the stimulation was observed (χ22 = 8.79, p = 0.012): participants were less likely to respond correctly under 7 Hz tACS stimulation both compared to the 18 Hz tACS stimulation (OR = 1.063; p = 0.019) and to the sham condition (OR = 0.949; p = 0.028). No difference in accuracy was found between the 18 Hz tACS stimulation and the sham condition (OR = 1.009; p = 0.698). Considering the interaction between discrimination and stimulation, no statistically significant effect was found (χ22 = 5.56, p = 0.062). A detailed breakdown of performance in all five trial types across stimulations is provided in the figures in the Supplemental Information: Fig. S2 illustrates the sham condition, Fig. S3 the seven Hz stimulation condition, and Fig. S4 the 18 Hz stimulation condition. Additionally, performance in illusory trials across all three stimulation conditions (seven Hz, 18 Hz, and sham), which provides a direct comparison of the effects of stimulation in illusory contexts, is depicted in Fig. 6.

Figure 5 Performance in all trial types across tasks.

Comparison of performance in numerical and continuous discrimination tasks across all five trial types: control trials and illusory tests. The x-axis indicates the trial types, and the y-axis shows the estimated probability of correct responses (i.e., selecting the larger target stimulus or numerosity in control trials and the stimulus or numerosity presented in the small context during illusory trials). Boxplots depict the median, interquartile ranges, and outliers, providing a detailed view of response accuracy.

Figure 6 Performance in illusory trials across stimulations.

Comparison of performance in numerical and continuous discrimination tasks considering only illusory trials across all three stimulations (18 hz, seven hz and sham). The x-axis indicates the task, and the y-axis shows the estimated probability of correct responses (i.e., selecting the stimulus or numerosity presented in the small context during illusory trials). Boxplots depict the median, interquartile ranges, and outliers, providing a detailed view of response accuracy.

Considering the trial type, a statistically significant effect was observed (χ24 = 3352.26, p < 0.001). As for the previous models, the accuracy in congruent trials was lower, compared to all the other types of trials (all p < 0.001). Conversely, in incongruent trials, the accuracy was higher compared to all the other trial types (all p < 0.001). In the large trials, participants were more likely to respond correctly compared to small trials (OR = 1.100; p < 0.001). In illusory trials, the accuracy was higher, compared to both large (OR = 0.833; p < 0.001) and small trials (OR = 0.757; p < 0.001).

The interaction between the discrimination and the trial type emerged as a statistically significant (χ24 = 1930.38, p < 0.001, Fig. 5). In the continuous quantity discrimination, all the previous differences among trial types were found. The only exception concerned the difference in accuracy between large and small trials, which appeared to be no longer statistically significant (OR = 1.044; p = 0.277, Fig. 5). In the numerical discrimination, the direction of some differences changed: contrary to the results of the trial type main effect, in the case of large trials, participants were more likely to respond correctly compared to both small (OR = 1.159; p < 0.001) and illusory trials (OR = 1.355; p < 0.001); finally, in the case of small trials, participants were more likely to respond correctly compared to illusory trials (OR = 1.169; p < 0.001). All the other comparisons were statistically significant and coherent with the previous main effects (see Table S2). No further statistically significant effects emerged (all p > 0.05).

Reaction times

Stimulus types were significantly associated to different reaction times means (χ24 = 127.12, p < 0.001, Fig. 7): participants were slowest in large trials, followed by small and congruent trials, then illusory trials, with the fastest responses in incongruent trials (all p < 0.01).

Figure 7 Comparison of the reaction times in all five types of trials.

The x-axis specifies the trial types, and the y-axis shows reaction times in milliseconds. Bars represent 95% confidence intervals, allowing for interpretation of variability and consistency in response times with the different type of stimuli.

Power analysis results

The results of the power analysis indicated that, with the specified parameters, a sample size of 45 subjects would lead, in a future study, to a predicted power of 83.20% (95% CI: [80.74–85.47]) for the test of the interaction between the stimulus and the stimulation. Instead, a sample size of 50 subjects would lead to a predicted power of 87.60% (95% CI: [85.40–89.58]; see Fig. 8 for the power curve).

Figure 8 Power curve showing the sample size simulation and the related power analysis.

Power curves showing the interaction effect between stimulus and stimulation as the sample size increases. The x-axis represents the sample size (40, 45, and 50), and the y-axis represents the power of detecting interaction effects. Bars indicate standard errors, providing insight into the robustness of the study’s statistical power.

Discussion

The question of whether human spatial and numerical abilities are processed by the same neuro-cognitive system has been widely discussed (Hayashi et al., 2013; Skagerlund, Karlsson & Träff, 2016; Walsh, 2003). Here, we tested this hypothesis in humans by observing (a) whether the Delboeuf illusory effect is of the same magnitude with both types of quantities in a behavioural task and (b) whether, using different tACS stimulation frequencies, the strength of the Delboeuf illusion appears similar and to the same extent in both types of quantities. Our results suggest that there is some level of overlap in how the Delboeuf perceptual bias influences spatial decisions and numerosity judgments, as indicated by the significant correlation between performances in the two tasks. However, this correlation alone is not sufficient to conclusively support the hypothesis of a shared cognitive mechanism.

Indeed, our findings reveal a significant disparity in human performance between continuous quantity and number discriminations, with participants demonstrating markedly superior proficiency in discriminating between areas compared to numbers. This outcome aligns with existing literature, which suggests that humans, as well as other non-human animals, possess a cognitive advantage for processing spatial information over numerical quantities (e.g., Gazzola, Vallortigara & Pellitteri-Rosa, 2018; Hubbard eyt al., 2005; Leibovich & Henik, 2014; Lucon-Xiccato et al., 2015). Nonetheless, conflicting evidence exists in the literature regarding the extent of the relationship between continuous and discrete quantity processing. Indeed, some studies have reported weak or non-significant correlations between performance in these tasks (e.g., Cappelletti et al., 2014; Dormal, Andres & Pesenti, 2008; Droit-Volet, Clément & Fayol, 2008), suggesting that the association between continuous and discrete quantity processing may vary across individuals or experimental conditions. In this regard, our analysis uncovered two noteworthy correlations that shed further light on the observed discrepancy. The first correlation highlights a positive relationship between participants’ ability to discriminate continuous quantities and their ability to discriminate between numerosities, resonating with previous studies (Burr & Ross, 2008; Gebuis & Reynvoet, 2012). The second correlation elucidates a connection between participants’ susceptibility to the Delboeuf illusion when presented with continuous quantities and their susceptibility to the same illusion when presented with numerosity arrays, which underscores the role of perceptual biases in influencing numerical judgments (Anobile et al., 2018; Dormal et al., 2018). Interestingly, while participants overall demonstrated a significant perception of the normal Delboeuf illusion, our initial statistical analyses did not reveal a similar effect with numerical stimuli, which may challenge the hypothesis of a single perceptual mechanism underlying both types of quantities. This observation aligns with the significant differences in performance we just reported, where participants showed better proficiency in continuous quantity discrimination compared to numerical discrimination. However, further analysis did reveal a significant correlation between the ability to discriminate continuous quantities and numerosities, as well as a correlation between susceptibility to the Delboeuf illusion for both types of quantities. While these findings suggest a potential shared mechanism for magnitude processing, it is important to interpret these correlations with caution. Indeed, the correlation coefficients observed for the control trials (0.37) and the illusory trials (0.34) were similar, raising questions about the strength and nature of the relationship. Given that illusory trials involve both perceptual biases and physical characteristics, one might expect a stronger correlation in these trials if the correlation reflected a generalized magnitude system. The comparable correlation coefficients suggest that the relationship between spatial and numerical processing may not be as straightforward as initially proposed. These findings underscore the complexity of magnitude processing and highlight the need for further investigation to better understand the underlying cognitive and perceptual mechanisms. Alternatively, these findings may be influenced by factors such as task difficulty and methodological issues, complicating our understanding of the potential commonality in processing mechanisms. For instance, a source of complexity can be the sample size used: while our findings provide preliminary insights, the relatively small sample size (34 participants) could have limited the robustness of the results, particularly in relation to the interactions between tACS conditions and task performance. Therefore, we conducted a sample size estimation to enhance more robust results for future research (as discussed in the following section of limitations).

It is noteworthy to consider the possibility that the numerical discrimination may have posed greater difficulty for participants compared to the continuous quantity discrimination. In our previous study (Santacà & Granziol, 2023), we revealed that humans perceive the numerical Delboeuf illusion, contrasting with our initial findings in the current study. This discrepancy could be attributed to differences in task procedures, stimuli presentation, or participant characteristics between the two studies. In comparison to the previous study, which was conducted online, allowing for flexibility in the devices and resolutions used, the current research was conducted in a controlled laboratory environment guaranteeing consistency in conditions such as monitor distance and absolute size of the stimuli. Moreover, participants in this study were presented with the stimuli for a shorter duration of 150 ms per trial to avoid eye movement, in contrast to the 1,500 ms in the previous work (Santacà & Granziol, 2023). Lastly, the number of trials and overall duration of the experiment differed significantly between the two studies. While the previous study lasted approximately 15 min with a total of 120 trials (Santacà & Granziol, 2023), including 24 illusory trials, our study comprised 600 trials in total, with 60 trials consisting of the numerical Delboeuf illusion and 60 of the classic illusion. These differences in experimental setup and protocol may have contributed to variations in participants’ fatigue, attentional allocation, and overall task performance, warranting careful consideration in interpreting and comparing the results across the two studies. Therefore, future investigations should carefully consider task design and methodological aspects to better understand the factors contributing to variations in perceptual judgments across different quantity discrimination tasks.

Together with evaluating whether the Delboeuf illusion similarly influences spatial and numerical quantities in a behavioral task, we also investigated the impact of different tACS frequencies on illusion strength. The lack of a significant interaction between tACS and the type of discrimination task suggests that the modulation of illusion strength by tACS does not significantly differ between spatial and numerical quantities. If distinct mechanisms were at play for processing spatial versus numerical information, we would expect tACS to have different effects on illusion strength in these tasks, leading to a significant interaction. Therefore, the absence of a significant interaction between tACS and the type of discrimination task suggests that the underlying perceptual mechanisms governing the Delboeuf illusion might indeed operate within a common framework for processing magnitudes. This is consistent with Walsh’s theory (Walsh, 2003), which states that spatial, numerical, and temporal processing share a common metric that could be modulated by neural oscillations, as our tACS results imply. However, this result alone should not be interpreted as definitive evidence of a single underlying perceptual mechanism. The speculations on how to interpret this non-significant effect may be several and should be taken with caution (given the ambiguity of interpreting null effects). It is possible that tACS affects different cognitive processes in the two tasks, and further research is needed to explore these differences. An additional result of interest emerged when taking into consideration the effect of the different tACS on magnitude processing. We hypothesized that theta-frequency tACS over the right parietal cortex would enhance visual integration, strengthening the illusion. Conversely, beta-frequency tACS would reduce illusion strength by increasing visual segregation. Participants underwent three tACS sessions (seven Hz, 18 Hz, and no stimulation), performing quantity discrimination tasks involving classic and numerical Delboeuf illusions. In contrast to our initial hypothesis, the application of tACS at seven Hz targeting the parietal areas appears to diminish the strength of the perceptual illusion compared to 18 Hz stimulation. Our hypothesis was predicated on prior research by Stonkus et al. (2016), which demonstrated that theta-frequency tACS applied to parietal regions enhances processes related to perceptual integration. Specifically, Stonkus et al. (2016) observed improvements in perceptual integration tasks wherein participants were required to discern a target stimulus (a snake made of Gabor patches with similar orientations) amidst distractors (Gabor patches of varied orientations). We interpreted these findings as suggestive that theta tACS might augment the spatial integration necessary for resolving the Delboeuf illusion, wherein the perception of two identically sized stimuli differs depending on the size of surrounding context elements. Contrary to expectations, our study revealed that theta tACS at seven Hz attenuated the strength of the illusion compared to the 18 Hz stimulation condition. The precise mechanism underlying this unexpected outcome remains unclear. One plausible speculation is that 18 Hz increases the visual segregating mechanism only when distractors are actually recognized as such but, in this study, the context might be considered neutral by the participants. On the other hand, seven Hz tACS may potentiate mechanisms involved in inhibiting irrelevant information such as distractors, rather than enhancing perceptual integration processes. This conjecture might also explain the results reported by Stonkus et al. (2016), where participants were tasked with both integrating target stimuli and excluding distractors. In our experimental setup, although unrelated to the primary task, the background context could be construed as a distractor to inhibit. Nonetheless, such explanations remain speculative at this juncture. Notably, the application of parietal theta tACS was also associated with a decline in performance on visual memory tasks. This finding suggests the possibility that 7 Hz parietal tACS may exert its effects not only on perceptual mechanisms but also on working memory processes, potentially impeding the accurate encoding of information into working memory (Wolinski, Sauseng & Romei, 2018). The present study, considering its preliminary nature, has some limits that have to be mentioned. First of all, on the small sample size, despite it is common to have small cardinalities in studies involving physiological measures such as tACS, more participants could provide clearer suggestions on the results. Moreover, considering the exploratory nature of our study and the lack of prior research on this specific topic, it was not feasible to conduct a priori sample size analysis that could refer to previous evidence or specific context/design. The only available reference (Santacà & Granziol, 2023) did not include the tACS manipulation, central to our research. Nonetheless, we aligned our approach with this previous study to the extent possible, as it represented the most relevant source of information. To address this limitation, especially for future studies, we implemented a power analysis based on the estimates we found. Indeed, we examined the number of participants and the power magnitude needed to re-estimate some important effects we observed, such as the interaction between stimulus type and stimulation. Our result suggests that, in a future study, 50 participants could provide more stable results with adequate power. Secondly, on the interpretation of non-significant results with low sample and cross-sectional data, it has been long debated that it is difficult to support the absence of an effect as a proof of the real nonexistence of it (Cohen, 1994). It is important to stress that the result provided by the present work should be read coherently with this position.

In humans, the Delboeuf illusion is thought to arise from a mix of assimilation and contrast effects (King, 1988). Specifically, in the traditional form of the illusion, when the target circle is positioned closer to the surrounding circle’s boundary, it tends to appear larger due to assimilation. Conversely, when the target circle is farther from the surrounding circle, it is perceived as smaller; an effect attributed to contrast. To explore how the background influences the perception of targets with actual physical differences, we designed both congruent and incongruent control trials. Our findings, consistent with previous research, show that participants are notably less precise in estimating target sizes in congruent trials. This is likely because, in such configurations, the smaller target circle tends to be assimilated into the smaller surrounding circle, while the larger target circle contrasts with the larger surrounding circle, leading to a reduced perceived difference in their sizes. In incongruent trials, the opposite occurs: the smaller target circle appears even smaller when contrasted with a larger surrounding circle, and the larger target circle seems even larger due to assimilation with a smaller surrounding circle. This effect amplifies the perceived size difference between the targets in incongruent trials, altering the accuracy of size estimations compared to congruent trials. Interestingly, a similar pattern of significantly enhanced performance in incongruent trials has been observed in numerical discrimination tasks. This finding suggests that the mechanisms underlying the perception of both continuous and discrete quantities may share commonalities, supporting the idea of a single magnitude system. However, it is important to note that while the performance in incongruent trials indicates a greater accuracy in both types of quantity discrimination, the average accuracy scores differ between continuous and numerical quantities, highlighting the need for further investigation.

So overall, our study examined whether the Delboeuf illusion, both in its classical and numerical forms, supports the hypothesis of a shared magnitude system between spatial and numerical quantities. This issue is particularly interesting given recent insights by Lourenco & Aulet (2023), who have provided a comprehensive framework for understanding how similarities between different magnitudes, such as physical size and numerosity, might arise from shared mechanisms. Indeed, these similarities can be accounted for by distinguishing between perceptual interactions, which occur at the level of sensory processing, and cognitive interactions, which involve higher-order cognitive processes. Specifically, Lourenco & Aulet (2023) suggest that perceptual interactions are likely to reflect the operation of a generalized magnitude system that is responsible for the initial encoding of magnitudes across different dimensions. This system could explain why certain illusions, such as the Delboeuf illusion, affect both spatial and numerical judgments in similar ways. In contrast, cognitive interactions refer to the more abstract and conceptual processing that might occur after the initial perceptual encoding, where different dimensions might influence each other through more complex cognitive mechanisms. Our findings reveal a significant correlation between participants’ ability to discriminate continuous quantities and numerosity, which could be interpreted as evidence of the perceptual interactions described by Lourenco & Aulet (2023). However, the differences in accuracy between the two discrimination tasks suggest that while there may be a shared perceptual basis, the cognitive processes involved in evaluating spatial and numerical information might diverge at higher levels of processing. This divergence aligns with the notion of cognitive interactions, in which different types of magnitude judgments are influenced by distinct cognitive processes. For instance, Lourenco & Aulet (2023) highlight how perceptual illusions, such as those involving size and number, can be influenced not only by the generalized magnitude system at the perceptual level but also by higher-order cognitive processes, including attention, working memory, and decision-making. They propose that cognitive interactions might arise when different dimensions, like size and numerosity, are processed together, leading to complex interdependencies that can modulate the final perceptual experience. For example, when a task requires simultaneous processing of spatial and numerical information, cognitive resources such as attention may need to be divided, potentially leading to interference effects that can alter the perception of these magnitudes. This perspective offers a potential explanation for the variability in how the Delboeuf illusion affects spatial versus numerical judgments observed in our study. While the perceptual interactions might account for the initial similarity in how the illusion influences both types of judgments, the subsequent cognitive interactions could introduce discrepancies depending on how cognitive resources are allocated during the task. In our study, participants demonstrated a significant perception of the classical Delboeuf illusion in the continuous quantity discrimination, suggesting that perceptual processes were strongly engaged. However, the numerical Delboeuf illusion was not as consistently perceived, which could be indicative of the additional cognitive demands associated with numerical processing, such as the need for precise counting or estimation under time constraints. Furthermore, how much these cognitive interactions influence perception might vary depending on individual differences in cognitive capacity, such as working memory span or attentional control. This variability could explain why some participants in our study showed strong susceptibility to the numerical Delboeuf illusion, while others did not, despite being subjected to the same perceptual conditions. It also suggests that future research could benefit from exploring how individual cognitive differences interact with perceptual processes in the context of magnitude illusions, potentially using neuroimaging or electrophysiological methods to uncover the underlying neural correlates.

Conclusions

By combining behavioral tasks with transcranial alternating current stimulation (tACS), our study provides new insights into the perceptual mechanisms underlying the Delboeuf illusion. Participants demonstrated superior performance in continuous quantity discrimination compared to numerical discrimination, though no significant interaction with tACS was observed. This supports the theory of a shared perceptual mechanism for time, space, and quantity, as proposed by Walsh (2003), while also highlighting potential cognitive distinctions in numerosity processing. The observed differences between continuous and numerical discrimination suggest additional cognitive demands associated with numerical tasks, possibly involving higher-order functions such as attention and working memory. These findings align with Harvey et al. (2015), who reported a topographic organization in the parietal cortex for numerosity, but also emphasized the hierarchical complexity of the generalized magnitude system. Indeed, the higher performance observed in continuous quantity discrimination might indicate that, while these dimensions share a common neural architecture, the processing of numerosity could involve additional or distinct cognitive pathways, reflecting the hierarchical nature of the generalized magnitude system. Notably, the inconsistent replication of the numerical Delboeuf illusion compared to prior studies (Santacà & Granziol, 2023) underscores the need to investigate how task design, stimulus presentation, and individual cognitive factors influence these processes. Future research should explore the neural correlates of numerical perception using techniques such as neuroimaging and electrophysiology while considering methodological refinements like eye-tracking and tailored task parameters to address variability in performance. These findings contribute to a deeper understanding of the interaction between perceptual and cognitive systems in magnitude processing, emphasizing the intricate relationships between spatial and numerical perception and their ecological and neural underpinnings. Indeed, while there is some evidence supporting a connection between spatial and numerical quantity processing, the inconsistencies observed suggest that the interaction between these processes may be more intricate than previously thought.

Supplemental Information

Supplemental Information 1 Raw individual data

Supplemental Information 2 Supplementary material

The electric field distribution, individual and group performance, additional figures of the performances in the two discrimination tasks and all the post-hoc comparisons of all the GLMMs.

The authors are thankful to Akin Donmez for the help in collecting the data and all the students for their participation in the study. This work was carried out within the scope of the project “use-inspired basic research” for which the Department of General Psychology of the University of Padova has been recognized as “Dipartimento di eccellenza” from “Ministero dell’Istruzione, Università e Ricerca” (MIUR, Italy).

Additional Information and Declarations

Competing Interests

Author Contributions

Human Ethics

Data Availability

The authors declare there are no competing interests.

Maria Santacà conceived and designed the experiments, prepared figures and/or tables, authored or reviewed drafts of the article, and approved the final draft.

Francesco Carabba analyzed the data, prepared figures and/or tables, and approved the final draft.

Achini Mihindukulasooriya Fernando performed the experiments, prepared figures and/or tables, and approved the final draft.

Enrico Pitocco performed the experiments, prepared figures and/or tables, and approved the final draft.

Luca Battaglini conceived and designed the experiments, prepared figures and/or tables, authored or reviewed drafts of the article, and approved the final draft.

Umberto Granziol conceived and designed the experiments, analyzed the data, prepared figures and/or tables, authored or reviewed drafts of the article, and approved the final draft.

The following information was supplied relating to ethical approvals (i.e., approving body and any reference numbers):

The study was approved on the 11th of April 2023 by the ethics committee (Protocol no. 5179) of the Department of General Psychology at the University of Padova (Italy).

The following information was supplied regarding data availability:

All raw individual data and statistics are available in the Supplemental File.

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
