# Peer review of "Exploring the classical and numerical Delboeuf illusion: the impact of transcranial alternating current stimulation on magnitude processing"

_PeerJ, doi:10.7717/peerj.19064_

## Round 0.1 · original submission · Minor Revisions

Two reviewers have assessed your manuscript and both provided some relatively minor suggestions that might improve your work. I think that these are quite straightforward and do not need any further comment from my side. Please reply to each point raised by the reviewers and implement the changes that you consider useful.

Reviewer 1 ·

Basic reporting

This study investigated the relationship between spatial and numerical magnitude processing using the Delboeuf illusion and transcranial alternating current stimulation (tACS). Thirty-four participants performed numerical and continuous quantity discrimination tasks under three stimulation conditions (7 Hz, 18 Hz, sham). Results showed higher accuracy in continuous quantity versus numerical tasks, with 7 Hz tACS unexpectedly reducing performance compared to other conditions. Significant correlations between tasks suggest shared mechanisms for spatial and numerical processing. The findings contribute to understanding how different types of quantities are processed in human perception. The English language usage is clear and professional throughout. The manuscript has clear research objectives and hypotheses. While the basic finding is interesting, there are some concerns that should be addressed.

Experimental design

No comment.

Validity of the findings

1. As the study aimed to investigate the existence of a generalized magnitude system responsible for processing different types of magnitudes (spatial vs. numerical), it is crucial to provide direct comparisons of spatial and numerical Delboeuf illusions (i.e., illusory trials) under each stimulation condition (7 Hz, 18 Hz, placebo). These comparisons are essential for addressing the study's main objective and should be accompanied by a graphical demonstration of the results.
2. The correlation between spatial and numerical performance is intriguing but requires a more cautious interpretation regarding causality. The correlation could also be attributed to the influence of a third factor, such as attention or response bias. Moreover, the comparable correlation coefficients observed for control trials (0.37) and illusory trials (0.34) raise questions about the nature of the correlation. If the correlation reflected a generalized magnitude system at the perceptual or illusory level, one would expect larger coefficients for the illusory trials compared to the control trials, as the illusory trials involve both perceptual and physical characteristics.
3. The authors should provide a sample size justification in the methods section to ensure the study's statistical power and reproducibility.
4. The rationale for selecting specific tACS frequencies (7 Hz and 18 Hz) should be elaborated upon in the manuscript. The authors are advised to include relevant literature or theoretical frameworks that support the choice of these particular frequencies to strengthen the study's methodological foundation.

Additional comments

1. Figure legends could be more detailed to help readers interpret the results independently.

Reviewer 2 ·

Basic reporting

The manuscript is generally well-written, with professional and clear language throughout. The introduction provides a solid background and places the study in the context of the broader literature. The references are comprehensive and appropriate, covering key studies in magnitude processing and the Delboeuf illusion.

Experimental design

The research addresses an interesting and meaningful question: whether the classical and numerical Delboeuf illusions are processed by shared mechanisms, and how transcranial alternating current stimulation (tACS) influences these processes. The experimental design is rigorous, with well-controlled variables (e.g., controlling for cumulative surface area in numerical tasks). The methods are described in sufficient detail to allow replication. Ethical considerations are appropriately addressed. The power analysis is a helpful addition, but it should be mentioned earlier in the discussion to contextualize the limitations.

Validity of the findings

The findings provide partial support for the hypothesis of a shared mechanism for spatial and numerical magnitude processing. The significant correlation between performance in the two tasks is compelling but not conclusive, given the differences in accuracy between continuous and numerical discriminations. The unexpected reduction of the illusion under 7 Hz tACS warrants further investigation, as the explanation provided (e.g., inhibitory mechanisms) is speculative. Statistical analyses are therefore well conducted.

Additional comments

I think that all the results of the tACS should be reported in a figure. This would help the reader quickly grasp the effects of tACS on the illusions.

Why is the power analysis not reported at the beginning of the Method section?

The Conclusions are too long. Please reduce the lenght of this section.

When discussing the study by Picon et al., in my opinion, such studies are not very conclusive in determining whether the Ebbinghaus illusion context actually affects numerosity perception. This is because perfectly controlling low-level features in numerosity perception is very challenging, and in this study, numerosity remains confounded with low-level features. It would be helpful to also reference the study by Adriano, Girelli, & Rinaldi (2022), Psychonomic Bulletin & Review, 29(1), 123–133. This study supports the hypothesis of a general mechanism for processing both discrete (e.g., number) and continuous dimensions (e.g., space) in the parietal areas (e.g., Walsh, 2003). However, they used the Ponzo illusion and the connectedness illusion applied to the same stimulus, demonstrating that numerosity perception is influenced by spatial illusions while controlling all low-level features using IC connections. This evidence shows that numerosity processing relies on discrete items, independently of spatial dimensions, and that numerosity perception is affected by spatial illusion (Ponzo illusion).

---

## Round 0.2 · accepted · Accept

The authors have addressed all of the reviewers' comments and the manuscript is now ready for publication.

Reviewer 1 ·

Basic reporting

The authors answered my questions well. I have no other comments.

Experimental design

No comment.

Validity of the findings

No comment.

Additional comments

No comment.

Reviewer 2 ·

Basic reporting

no comment

Experimental design

no comment

Validity of the findings

no comment

Additional comments

The authors have incorporated the requested revisions into the paper and have addressed the concerns that were raised.